# Real-IKEA: Why Physical Fidelity is the Prerequisite for Robust Manipulation

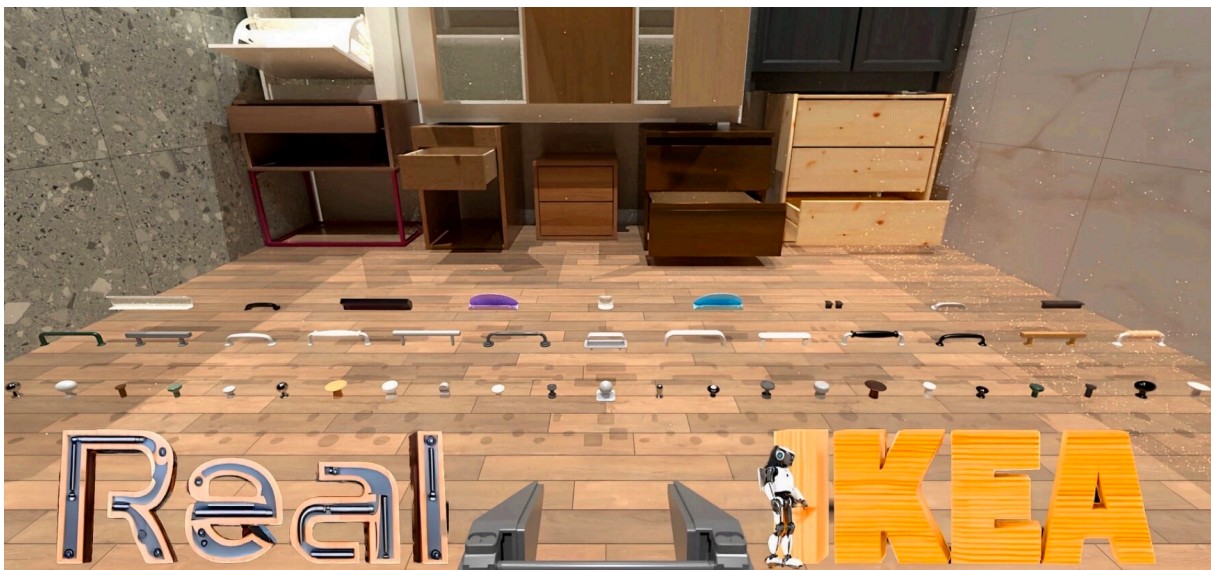

Fig. 1: **Overview of the Real-IKEA.** Real-IKEA provides a large-scale, high-fidelity library of articulated configurations as the foundation for learning robust, contact-rich manipulation policies.

*Abstract*— **Robotic manipulation robustness often founders on the physics gap between simplified simulations and the resistance-laden real world. In this work, we emphasize that *physical realism* in articulated interaction is an important ingredient for robust policy learning. We present Real-IKEA, a dataset and simulation framework designed with physical accuracy as a first-class goal. Real-IKEA provides 1,079 articulated asset configurations, derived from 83 authentic IKEA handles and knobs processed through a meticulous six-step physical workflow. For contact-geometry accuracy, we introduce a bidirectional surface-deviation metric ($E_{Q \to P}$, $E_{P \to Q}$) to quantify collision meshes. For dynamics realism, we establish resistance-calibrated configurations that vary damping and friction. Crucially, we demonstrate through a Reinforcement Learning (RL) policy that high-fidelity assets enable the discovery of robust "hooking" and "levering" strategies that prioritize mechanical advantage over fragile friction-pulling. Together, these results position Real-IKEA as a critical benchmark for developing manipulation policies capable of human-level robustness in articulated object tasks.**

## I. INTRODUCTION

Learning manipulation policies for robots has long relied on simulation as a scalable and cost-effective source of data [1]–[5]. The *data pyramid* paradigm illustrates this strategy: massive synthetic data forms the base, supplemented by smaller amounts of curated real-world data at higher tiers. While effective in locomotion and navigation [6]–[9], robust behavior in *contact-rich manipulation tasks* remains hard to obtain [10]–[12]. Beyond visual domain gaps, many failures are fundamentally mechanical: contact location, leverage, and resistance dominate outcomes.

This paper highlights an under-emphasized point: *physical fidelity is critical for robust policy learning*. We focus on two physical factors directly shaping policy discovery: (1) contact geometry fidelity of interactive parts like handles and knobs, and (2) realistic damping and friction in articulated joints.

Despite abundant articulated object benchmarks, physical fidelity remains a bottleneck. Datasets like PartNet-Mobility [1], [2] feature synthetic assets diverging from real-world distributions, where oversimplified handles introduce out-of-distribution (OOD) risks for modern domestic environments [13], [14]. While Adamanip [15] and ManiSkill [3], [16] improve visual quality and task diversity, their assets remain heavily restricted by simplified CAD models and coarse collision meshes.

The effectiveness of contact-rich tasks in simulation depends heavily on how the physics engine handles the *contact manifold*. Mainstream simulators [17]–[19] rely on convex meshes for collision handling. When these are coarse, non-convex features—like holes in a handle or fine grooves in a knob—are treated as solid convex volumes, effectively "sealing off" the potential interaction space. This geometric

discrepancy reveals a fundamental flaw in datasets assuming collision meshes equal their visual counterparts [1], [2], [15]. Actual collision geometry is often a distorted, "swollen" version of the object. This precludes realistic contact-rich research, as the strategies explored do not reflect real-world physics. This limitation has remained largely hidden because contemporary policies often default to simplified parallel-jaw grasps with fixed orientations followed by a linear pull.

However, these friction-dominated strategies are inherently fragile. Relying exclusively on a fixed-orientation grasp provides driving force entirely via surface friction, guaranteeing neither force nor form closure. Consequently, under real-world mechanical resistance (e.g., heavy drawers) or low-friction materials, these grasps inevitably slip and fail.

In contrast, human-level robustness stems from exploiting object geometry. When opening a stubborn drawer, humans do not simply pinch the handle and pull relying on skin friction; instead, we insert fingers into handle gaps (hooking) or wrap them around a knob's curved back. These geometry-exploiting behaviors create mechanical interlocking (form closure), providing immense resistance against slipping. This mechanical leverage is the fundamental source of robustness. Achieving this in robotics strictly requires simulation environments with meticulously accurate geometric and physical properties.

Motivated by this, we introduce **Real-IKEA**, a comprehensive framework designed to establish physical accuracy as a first-class goal for robust policy learning. Our contributions are three-fold: (1) **The Real-IKEA Physical Dataset**: 1,079 high-fidelity articulated assets combinatorially paired from authentic IKEA products, strictly preserving real-world geometric distributions. (2) **High-Fidelity Contact Modeling**: We replace standard coarse convex hulls with precise convex decompositions (quantified by our bidirectional surface deviation metric), re-enabling critical affordances like holes and grooves, alongside resistance-calibrated joint dynamics. (3) **Validation of Geometry-Exploiting Robustness**: We provide empirical proof, via an RL policy, that physical fidelity is the prerequisite for robustness. We demonstrate that high-fidelity assets uniquely enable the emergence of form-closure strategies (e.g., hooking) that succeed under real-world challenges where friction-dominated baselines collapse.

## II. THE REAL-IKEA PHYSICAL FRAMEWORK

TABLE I: Feature comparison across articulated-object datasets. Real-IKEA is uniquely positioned to support robustness research by combining precise collision modeling with calibrated joint dynamics.

| Environment | Reusable Interactive Parts | Accurate Collisions (Decomp.) | Configurable Joint Resistance (Damp./Fric.) | Real-Object Digital Twins |
|---|---|---|---|---|
| SAPIEN | ✗ | ✗ | ✗ | ✗ |
| AdaManip | ✓ | ✗ | ✗ | ✗ |
| UniDoorManip | ✓ | ✗ | ✗ | ✗ |
| **Real-IKEA** | ✓ | ✓ | ✓ | ✓ |

### A. Dataset Construction: Grounding Assets in Reality

A central challenge in building an articulated object dataset is ensuring the geometric distribution accurately reflects the real world. Rather than synthesizing arbitrary shapes, we turn to IKEA products, which provide a globally standardized yet diverse set of domestic furniture. We adopt IKEA cabinets as canonical base units to ensure coverage of all common joint mechanisms. Crucially, IKEA's design philosophy emphasizes modularity—handles and knobs are manufactured as interchangeable units.

Leveraging this modularity, we curated a comprehensive library of 83 authentic IKEA handles and knobs. By systematically combining them with base cabinets, we generated **1,079** articulated configurations. As shown in Table I, Real-IKEA fills a critical void: while frameworks like AdaManip scale via part reuse, their operable components remain morphologically simplified and physically coarse. Real-IKEA provides a large-scale, morphologically diverse library whose scales, dimensions, and structural combinations are rigorously aligned with physical reality.

### B. Closing the Contact-Geometry Gap

While high-resolution visual meshes provide the necessary foundation for accurate geometric decomposition, they do not automatically yield realistic physical interactions. The core limitation in modern simulators is the handling of non-convex geometries. In prior datasets, collision meshes are often crudely approximated by simple convex hulls. Despite its ubiquity, the magnitude of this approximation error—and its catastrophic impact on learning robust manipulation—has been largely ignored.

To resolve this, we process our collision meshes using the COACD [20] algorithm, which produces hundreds of high-fidelity convex primitives per asset. To rigorously quantify this improvement, we introduce a metric to evaluate collision mesh accuracy. We uniformly sample dense surface points on both the ground-truth visual mesh (*standard shell*, $P$) and the collision mesh (*collision shell*, $Q$). For each point $q \in Q$, we compute its nearest-neighbor distance to $P$, measuring the outward deviation (i.e., "mesh swelling") of the collision boundary:

$$E_{Q \to P} = \frac{1}{|Q|} \sum_{q \in Q} \text{dist}(q, P), \qquad \text{dist}(q, P) := \min_{p \in P} \|q - p\|_2.$$

Symmetrically, we compute $E_{P \to Q}$ to yield a bidirectional measure of geometric discrepancy.

As illustrated in Figures 2 and 3, baseline collision meshes exhibit massive deviations. Crucially, these deviations occur exactly in the most mechanically vital regions: the inner loops of handles, deep grooves, and curved support structures. In standard simulators, these regions are effectively "filled in," blocking the robot's end-effector from entering the space. By contrast, our $E_{Q \to P}$ optimized meshes preserve these cavities. This physical fidelity is mandatory for robust manipulation, as it re-enables the affordances required for a robot to achieve form-closure (e.g., hooking a finger completely through a handle).

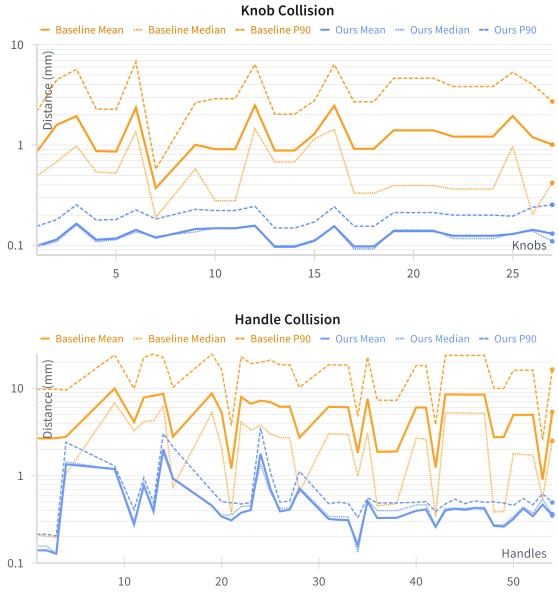

Fig. 2: Evaluation of physical interaction fidelity. We visualize the statistical distribution (Mean, Median, and P90) of the point-wise distances $\text{dist}(q,P)$ for Real-IKEA interactive assets. Note that the mean metric corresponds to $E_{Q\to P}$ and $E_{Q'\to P}$, where $Q'$ denotes the baseline collision shell. Our rigorously processed meshes achieve significantly lower geometric swelling across all metrics.

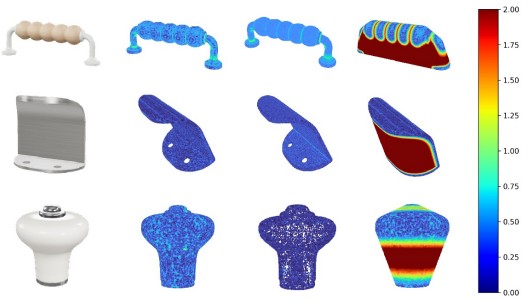

Fig. 3: Heatmap visualization of collision errors. Left to right: (1) visual mesh $P$, (2) our error $\text{dist}(p,Q)$, (3) our bidirectional error $\text{dist}(q,P)$, and (4) baseline error $\text{dist}(q',P)$. Red indicates severe deviation. Crucially, baseline approximations ($Q'$, right) completely obscure functional cavities like handle holes.

## III. EVALUATING ROBUSTNESS: THE FRICTION TRAP VS. GEOMETRY EXPLOITATION

Conventional robotic manipulation often relies on simple parallel-jaw grasps that fail to exploit object affordances. When faced with real-world mechanical resistance, these friction-dominated grasps inevitably slip. Overcoming this requires *contact-rich manipulation*, where the end-effector adapts its pose and utilizes contact geometry to establish reliable mechanical locks.

As shown in Figure 4, Real-IKEA assets reproduce real-world challenges and enable realistic strategies to overcome

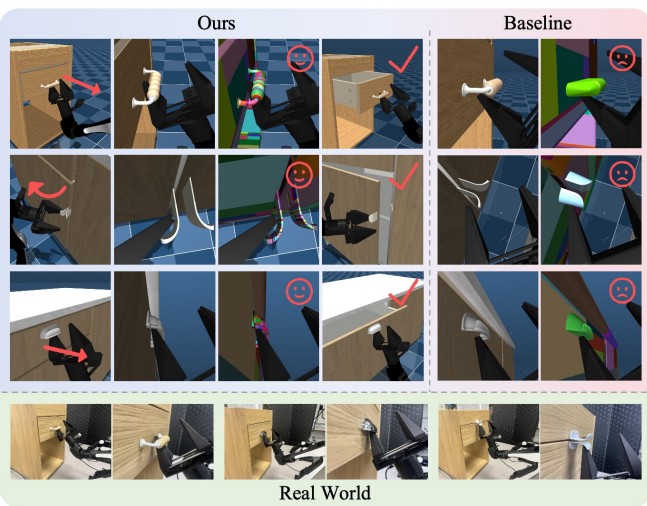

Fig. 4: Real-IKEA enables realistic contact-rich strategies that are physically impossible in coarse simulators. High-fidelity meshes allow agents to explore complex interaction modes such as hooking and leveraging.

them. Thanks to our precise collision modeling, the environment supports non-trivial actions such as *hooking* under a handle or *pushing laterally* against a knob. By contrast, conventional simulators—whose collision meshes are coarse and often omit holes or curved surfaces—completely "seal off" these interaction spaces, forcing robots into fragile, friction-based behaviors.

### A. Case Study: Drawer Opening Under Joint Resistance

We isolate a canonical task: opening a drawer with a parallel-jaw gripper under varying joint resistance. We configure three resistance modes to test robustness: *Smooth* (damping = 2, friction = 5), *Normal* (damping = 5, friction = 10), and *High Resistance* (damping = 10, friction = 20).

We compare three representative strategies: (1) **Human teleoperation** as an upper bound; (2) **Ground-truth baselines** using heuristic friction-based pulling; and (3) **Grasp-Gen** [21], a state-of-the-art model that predicts 6-DoF grasp poses.

### B. Results Across Handle Types and Implications

The results in Table II reveal a stark contrast between friction-dominated methods and geometry-aware manipulation.

**The Collapse of Friction-Based Strategies.** For knobs and finger-pull handles, both GraspGen and heuristic baselines perform well in the *Smooth* setting but collapse under high resistance (0% success). Relying purely on surface friction generated by pinching, these methods inevitably slip when the pulling force required to overcome joint resistance exceeds the frictional limit.

**Robustness via Geometry Exploitation.** In contrast, human teleoperation maintains high success, particularly on two-point handles (perfect success across all levels). Analysis shows that humans exploit *form-closure*: inserting a finger

TABLE II: Success rate distribution across different manipulation policies under varying joint resistances. F: Fail, P: Partial, S: Success.

(a) Knobs

| Policy | Smooth | | | Normal | | | High Fric. | | |
|---|---|---|---|---|---|---|---|---|---|
| | F | P | S | F | P | S | F | P | S |
| GraspGen | 0.24 | 0.07 | 0.69 | 0.76 | 0.21 | 0.03 | 1.00 | 0.00 | 0.00 |
| Normal Ways | 0.00 | 0.00 | 1.00 | 0.59 | 0.31 | 0.10 | 1.00 | 0.00 | 0.00 |
| Human Teleop | 0.00 | 0.00 | 1.00 | 0.03 | 0.17 | 0.79 | 0.10 | 0.52 | 0.38 |

(b) Finger-pull handles

| Policy | Smooth | | | Normal | | | High Fric. | | |
|---|---|---|---|---|---|---|---|---|---|
| | F | P | S | F | P | S | F | P | S |
| GraspGen | 0.31 | 0.00 | 0.69 | 0.92 | 0.08 | 0.00 | 1.00 | 0.00 | 0.00 |
| Normal Ways | 0.00 | 0.00 | 1.00 | 0.85 | 0.15 | 0.00 | 1.00 | 0.00 | 0.00 |
| Human Teleop | 0.00 | 0.00 | 1.00 | 0.00 | 0.04 | 0.96 | 0.00 | 0.65 | 0.35 |

(c) Two-point handles

| Policy | Smooth | | | Normal | | | High Fric. | | |
|---|---|---|---|---|---|---|---|---|---|
| | F | P | S | F | P | S | F | P | S |
| GraspGen | 0.04 | 0.00 | 0.96 | 0.61 | 0.07 | 0.32 | 0.89 | 0.00 | 0.11 |
| Normal Ways | 0.00 | 0.00 | 1.00 | 0.72 | 0.07 | 0.21 | 1.00 | 0.00 | 0.00 |
| Human Teleop | 0.00 | 0.00 | 1.00 | 0.00 | 0.00 | 1.00 | 0.00 | 0.00 | 1.00 |

through the loop or bracing against curvature. This strategy provides a mechanical lock that makes the driving force independent of friction.

**Conclusion on Policy Design.** Robust manipulation under resistance hinges on converting actuator effort into opening torque $\tau = r \times f$. This motivates a transition toward closed-loop, geometry-exploiting formulations—a paradigm we validate next via our RL framework.

## IV. DISCOVERING ROBUSTNESS: A REINFORCEMENT LEARNING PARADIGM

To empirically validate that our high-fidelity physical assets are the prerequisite for human-level manipulation robustness, we design a Reinforcement Learning (RL) framework. As illustrated in Figure 5, our goal is to demonstrate that when an agent is trained in an environment with accurate contact geometry and calibrated resistance (Real-IKEA), it naturally escapes the "friction trap" and discovers geometry-exploiting strategies.

### A. Policy Training via Privileged Information

As detailed in Figure 5, we train an RL policy using Proximal Policy Optimization (PPO) [22]. To maximize the policy's ability to explore complex physical interactions, the agent is provided with a low-dimensional privileged state observation, including precise handle coordinates, drawer joint states, and the 6D pose of the end-effector. The policy outputs a 5D continuous action vector ($\Delta x, \Delta y, \Delta z, \Delta$pitch, and gripper action), which is mapped to physical movements via an Inverse Kinematics (IK) solver. By properly scaling the translation and pitch commands, we ensure the agent can execute smooth, high-frequency closed-loop adjustments. Further training details, including our reward curriculum

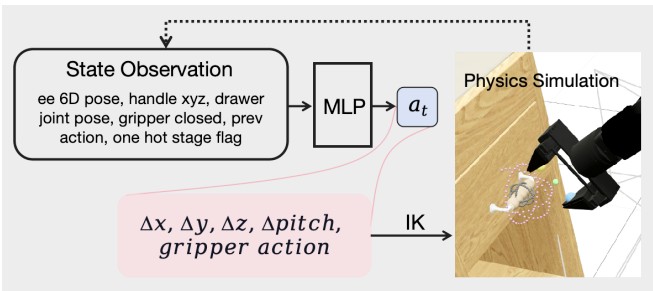

Fig. 5: Overview of the RL Policy. The agent observes a privileged low-dimensional state and outputs a 5D action to interact with the high-fidelity physics simulation, inducing robust, contact-rich manipulation.

and domain randomization schedules, are provided in the Appendix.

### B. Emergence of Form-Closure Strategies

The most critical outcome of this RL pipeline is the qualitative nature of the learned behaviors. When trained on standard, simplified collision meshes, the RL agent converges to naive, friction-based pulling—which inevitably fails under real-world resistance.

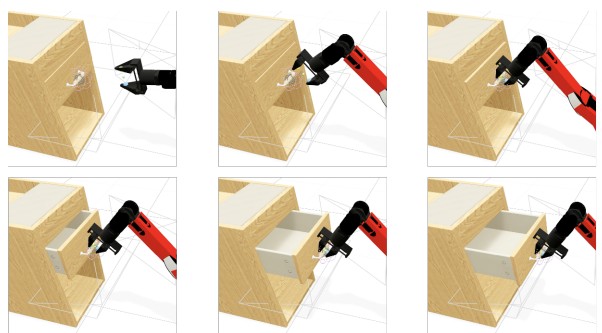

Fig. 6: Emergence of form-closure strategies. Trained on Real-IKEA assets, the policy discovers advanced mechanical interactions, such as hooking fingers entirely through handle loops to counter high joint resistance.

However, as visually confirmed in Figure 6, when trained on Real-IKEA assets, the policy discovers completely different local optima. It actively seeks *form-closure*: for two-point handles, the agent learns to thread its finger completely through the handle loop before pulling. These emergent, closed-loop strategies provide definitive proof that physical fidelity is the foundational driver of manipulation robustness.

## V. CONCLUSION

In this work, we introduced **Real-IKEA**, a dataset and simulation framework that systematically tackles the physics gap in articulated manipulation by establishing physical fidelity as a first-class goal. By providing **1,079** high-fidelity digital twins with precise collision meshes and calibrated joint resistance, we enable the study of contact-rich robust policies that were previously impossible.

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

APPENDIX

*A. Real-IKEA Dataset Construction and Physical Modeling Details*

While the main text focuses on the emergence of robust manipulation policies, this appendix provides extended details regarding the construction of the Real-IKEA dataset, the specific physical failure modes we address, and further quantitative evidence of physical fidelity.

*1) The Four Characteristic Failure Modes in Manipulation:* The primary motivation for developing Real-IKEA stems from observing how conventional, friction-dominated policies fail when deployed on real articulated objects. As illustrated in Figure 8, these failures typically manifest in four characteristic modes:

i) **Slip:** When joint resistance exceeds the frictional force provided by a parallel-jaw pinch, the gripper simply slides off the handle.

ii) **Narrow Clearance:** Handles mounted close to the cabinet face require precise finger insertion strategies (e.g., hooking) that coarse collision meshes prohibit.

iii) **Mutual Interference:** Complex geometries or nearby structures can cause unwanted collisions if the policy cannot exploit the object's form.

iv) **Specialized Designs:** Unique shapes (like deeply curved knobs) require specific wrist rotations and levering actions to generate opening torque.

Standard simulators often mask these issues by using low joint resistance or inaccurate collision boundaries. Real-IKEA enforces these physical constraints to ensure that learned policies must acquire robust, geometry-exploiting strategies to succeed.

*2) Dataset Construction Workflow:* Unlike existing datasets that often rely on synthetically generated or simplified CAD models [2], all interactive assets in Real-IKEA are derived from authentic IKEA products. Constructing a single ready-to-use, high-fidelity articulated asset requires a meticulous **six-step processing workflow**:

i) **Component Segmentation and Formatting:** We manually segment the overall visual mesh into distinct

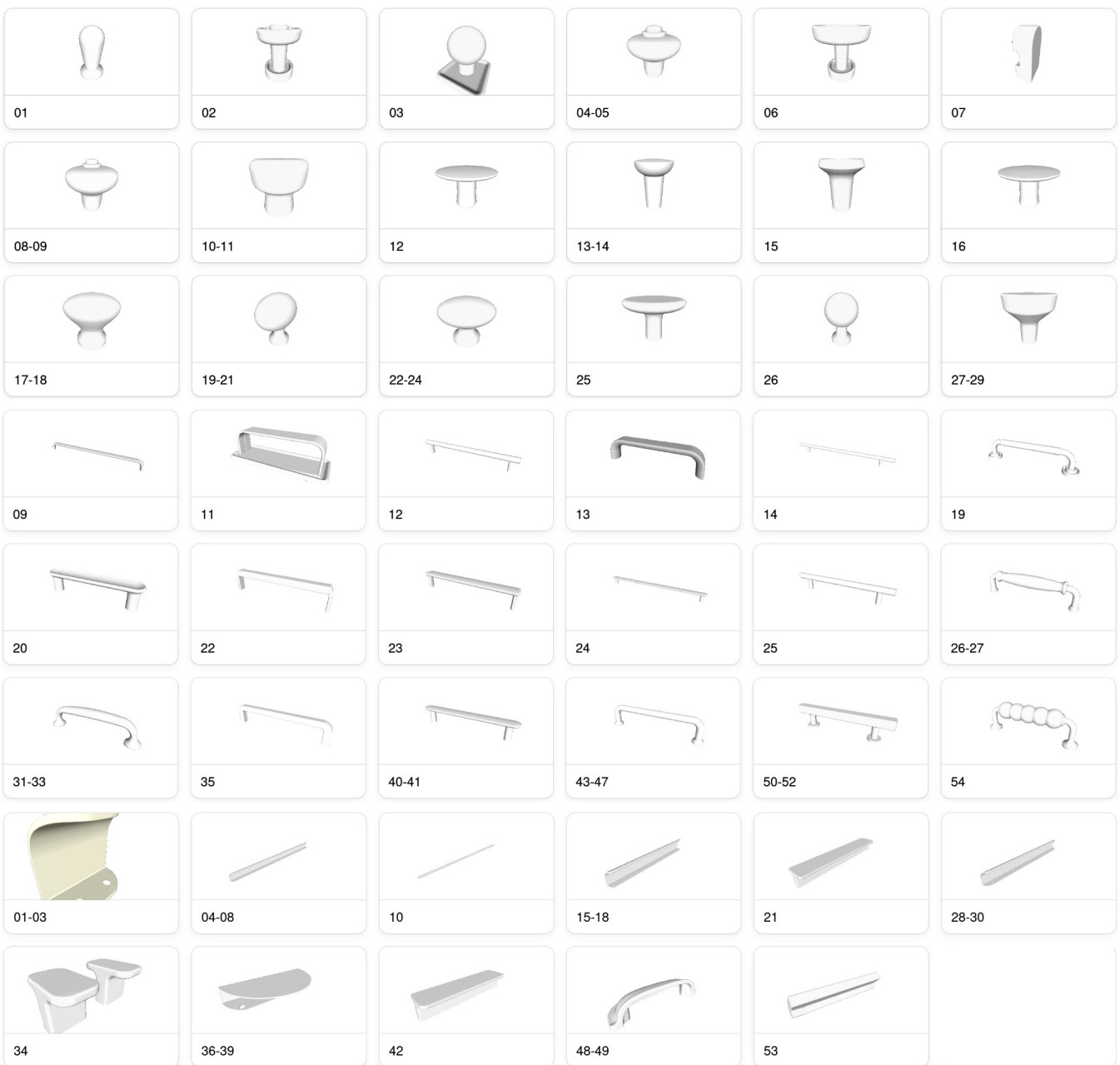

Fig. 7: The diversity of interactive components in Real-IKEA, categorized into knobs, two-point handles, and finger-pull handles.

components (e.g., cabinet base, drawers, doors, interactive parts). The initial mesh format is converted, and corresponding texture maps are generated.

ii) **Structural Cleaning for Simulation:** We manually refine the component meshes, removing internal features (like internal drawer rollers or complex tracks) that would cause problematic self-collisions in the simulator while preserving the external interaction surfaces.

iii) **Pre-Collision Scaling and Clipping:** Because Approximate Convex Decomposition (ACD) inevitably causes a slight "swelling" of the resulting primitives, we manually scale or clip specific joint regions. Without this

step, tightly fitted components (e.g., a flush drawer face) might become stuck against the cabinet frame after decomposition.

iv) **Collision Mesh Generation:** Using the cleaned non-convex visual meshes, we apply the COACD algorithm [20] to generate high-fidelity collision geometry. This process produces hundreds of sub-meshes (convex primitives) per asset.

v) **Assembly and Joint Parameterization:** We manually assemble the individual components, accurately determining joint positions, axes, and physically realistic ranges. Crucially, we calibrate joint damping

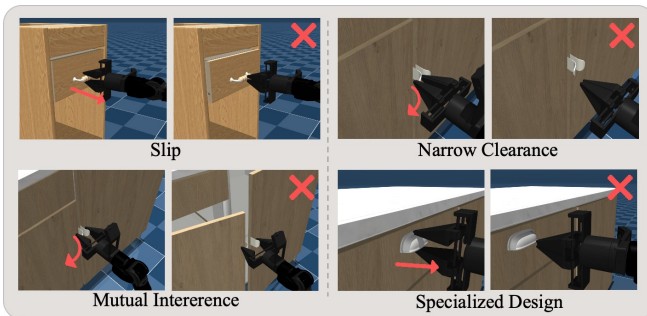

Fig. 8: Four characteristic failure modes observed when robots interact with real articulated objects. Real-IKEA is designed to faithfully reproduce these challenges in simulation.

and friction parameters to reflect real-world mechanical resistance.

vi) **Simulation Integration:** The visual meshes, accurate collision meshes, and joint kinematics are assembled into a unified XML format compatible with the MuJoCo simulator.

To ensure this rigorous process yields a high-quality foundation, Figure 9 details the geometric properties of our interactive parts compared to standard datasets. Our assets maintain significantly higher vertex and face counts, ensuring that fine geometric affordances are preserved before the ACD step.

*3) Interactive Component Diversity:* Leveraging IKEA's modular design philosophy, we combinatorially paired base cabinet units with a curated library of 83 authentic handles and knobs. As shown in Figure 7, we categorize these interactive parts into three main types based on their geometric affordances:

- **Knobs:** Require wrist rotation to wedge the gripper against curvature.
- **Two-point handles:** Permit full finger insertion (form-closure) through the loop.
- **Finger-pull handles:** Require hooking the fingertips under a narrow lip.

Furthermore, our dataset incorporates base furniture units that cover all major articulated joint types (Figure 10), facilitating broad evaluation of manipulation policies across diverse structures.

*4) Extended Collision Accuracy Visualization:* To visually confirm the quality of our COACD processing (Step 4 of our workflow), we provide a heatmap visualization in Figure 11. The metric $H_{Q \to P}$ represents the outward deviation from the collision shell to the visual shell. The results show that the collision models for knobs are highly precise. For complex handles, minor errors only appear in regions with extremely sharp curvature changes, while the critical interaction spaces (like the inner loops) remain open and physically accessible.

### B. Privileged RL Policy: Task, Observations, Actions, and Rewards

This appendix describes the RL policy training details.

*1) Simulation and Control Rate:* The simulator integrates rigid-body dynamics with a fixed physics timestep $\Delta t_{\text{sim}} = 0.002\,\text{s}$. A control decimation of $N_{\text{dec}} = 50$ yields a policy command interval

$$\Delta t = N_{\text{dec}}\,\Delta t_{\text{sim}} = 0.1\,\text{s}, \tag{1}$$

corresponding to a $10\,\text{Hz}$ low-level command rate. Episode duration is bounded by a maximum horizon (fixed wall-clock length in simulation time), after which a timeout terminates the episode.

*2) Privileged State:* The Policy observes low-dimensional features sufficient to specify the manipulation geometry: an end-effector–centric pose summary, the handle position in world coordinates, the drawer joint displacement, a gripper aperture–based closure indicator, the previous action (for temporal context), and a four-dimensional stage indicator aligned with the staged reward (one-hot over four stages).

*3) Action Parameterization and Kinematics:* The policy outputs a five-dimensional continuous action each control step, clipped to $[-1, 1]$ before execution. Four dimensions specify an incremental end-effector motion relative to the current configuration; one dimension commands the parallel gripper.

Let $(a_0, a_1, a_2, a_3)$ denote the clipped arm command. These are scaled to physical increments as

$$\delta_{\text{pitch}} = \alpha_{\text{pitch}}\,a_0, \qquad (\delta_x, \delta_y, \delta_z) = \alpha_{\text{trans}}\,(a_1, a_2, a_3), \tag{2}$$

with $\alpha_{\text{pitch}} = 0.04\,\text{rad}$ and $\alpha_{\text{trans}} = 0.01\,\text{m}$ per unit command. The controller composes a target rigid-body pose using fixed roll and yaw conventions tied to the nominal manipulator configuration, solves inverse kinematics for a feasible joint target, and tracks that target under joint limits. The gripper command is mapped through an affine law to a tendon actuator that opens or closes the jaws.

Inverse kinematics may fail on a step; such failures are penalized and, if persistent, can trigger early termination.

*4) Staged Reward: Geometry, Gates, and Monotonic Baselines:* Rewards are decomposed into four stages that encourage a coarse ordering: approach a pre-grasp region, refine approach to the handle neighborhood, adopt a grasp-ready closure, then open the drawer by increasing the slide displacement. Let $\mathbf{c}$ be the midpoint between the fingertips and $\mathbf{h}$ the handle position. Two approach targets are defined as $\mathbf{t}_1 = \mathbf{h} + \mathbf{o}_1$ and $\mathbf{t}_2 = \mathbf{h} + \mathbf{o}_2$ with fixed offsets $\mathbf{o}_1, \mathbf{o}_2$ (in our setting, both offsets share the same nominal lateral bias). Distances are $d_1 = \|\mathbf{c} - \mathbf{t}_1\|_2$ and $d_2 = \|\mathbf{c} - \mathbf{t}_2\|_2$. A small constant $\varepsilon = 10^{-2}$ stabilizes logarithmic shaping via $-\log(d + \varepsilon)$.

Let $w \in [0, 1]$ denote the normalized gripper aperture, where 0 and 1 represent a fully closed and fully open state, respectively. We define the geometric scores as binary indicator flags: $s_{\text{open}} = 1$ if $w > 0.6$ (and 0 otherwise), and $s_{\text{close}} = 1$ if $w < 0.4$ (and 0 otherwise). These flags provide discrete guidance to ensure the gripper maintains an appropriate configuration during the pre-grasp approach and the subsequent grasping phase. Stage transitions use distance thresholds $\tau_1$ and $\tau_2$ (we use $\tau_1 = 0.05$ and $\tau_2 = 0.02$) and

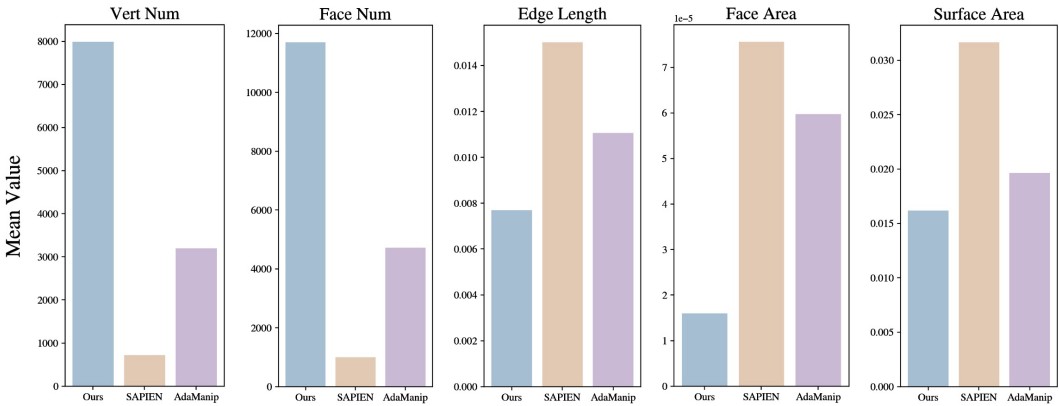

Fig. 9: Mesh statistics of the Real-IKEA interactive parts. Higher vertex counts and smaller face areas provide the high-resolution basis essential for robust collision modeling.

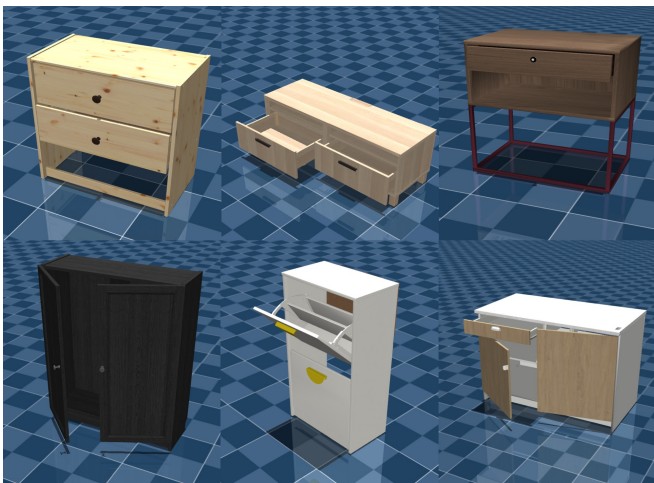

Fig. 10: Real-IKEA base assets encompass typical joint mechanisms (e.g., revolute doors, prismatic drawers) and support flexible composition with our interactive component library.

additional readiness conditions so that drawer opening is credited only after grasp readiness is achieved.

*a) Stage 1.:* The stage-one reward combines an open encouragement term with proximity to $\mathbf{t}_1$:

$$R_1^{\text{raw}} = w_1^{\text{open}} s_{\text{open}} + w_1^{\text{app}}\big(-\log(d_1+\varepsilon)\big), \qquad (3)$$

with $w_1^{\text{open}} = w_1^{\text{app}} = 2$.

*b) Stage 2.:* Stage two emphasizes refinement toward $\mathbf{t}_2$ without duplicating the stage-one "stay open" incentive: the open weight is set to zero ($w_2^{\text{open}} = 0$). Define per-stage saturation levels induced by the logarithmic approach term at $d \to 0$:

$$M_1 = w_1^{\text{open}} + w_1^{\text{app}}\log\frac{1}{\varepsilon}, \qquad (4)$$

$$M_2 = w_2^{\text{open}} + w_2^{\text{app}}\log\frac{1}{\varepsilon} = w_2^{\text{app}}\log\frac{1}{\varepsilon}, \qquad (5)$$

with $w_2^{\text{app}} = w_1^{\text{app}}$. The stage-two raw reward is baseline-shifted:

$$R_2^{\text{raw}} = M_1 + w_2^{\text{open}} s_{\text{open}} + w_2^{\text{app}}\big(-\log(d_2+\varepsilon)\big). \qquad (6)$$

This design intentionally removes an explicit open bonus in stage two so the policy can exploit geometric freedom to discover transitions toward closure in the next stage rather than being pinned to the same pre-grasp cue.

*c) Stage 3.:* Grasp readiness is encouraged by scaling the closure score:

$$R_3^{\text{raw}} = (M_1 + M_2) + \beta\, s_{\text{close}}, \qquad \beta = 10. \qquad (7)$$

*d) Stage 4.:* Let $q \geq 0$ denote the drawer slide displacement. Opening is rewarded by

$$r_{\text{open}}(q) = \kappa\log(100q+1), \qquad \kappa = 15. \qquad (8)$$

Near $q = 0$, the derivative is large, which prioritizes breaking static friction and producing an initial measurable motion—often the hardest part of manipulation. As $q$ grows, marginal incentive decreases. To preserve monotonicity across stage boundaries, the final stage adds a baseline offset equal to the maximum attainable contribution from earlier stages:

$$R_4^{\text{raw}} = r_{\text{open}}(q) + (M_1 + M_2 + \beta). \qquad (9)$$

At each step, only one stage contribution is active under mutually exclusive gating; the stage indicator provided to the policy matches this decomposition.

*5) Auxiliary Terms and Terminations:* A small penalty discourages jerk; inverse-kinematics failures incur a per-step cost. Episodes may also terminate on timeout, prolonged kinematic infeasibility, or stagnation under commanded motion (task-specific definitions).

*6) Domain Randomization and Two-Phase Training:* Training proceeds in two phases. First, the handle and robot base poses are fixed at reset to reduce variance while acquiring a feasible skill. Second, per-episode pose perturbations are applied to the handle and base within bounded ranges to improve robustness. Specifically, the switch to the second phase occurs after 900 policy iterations, at which point

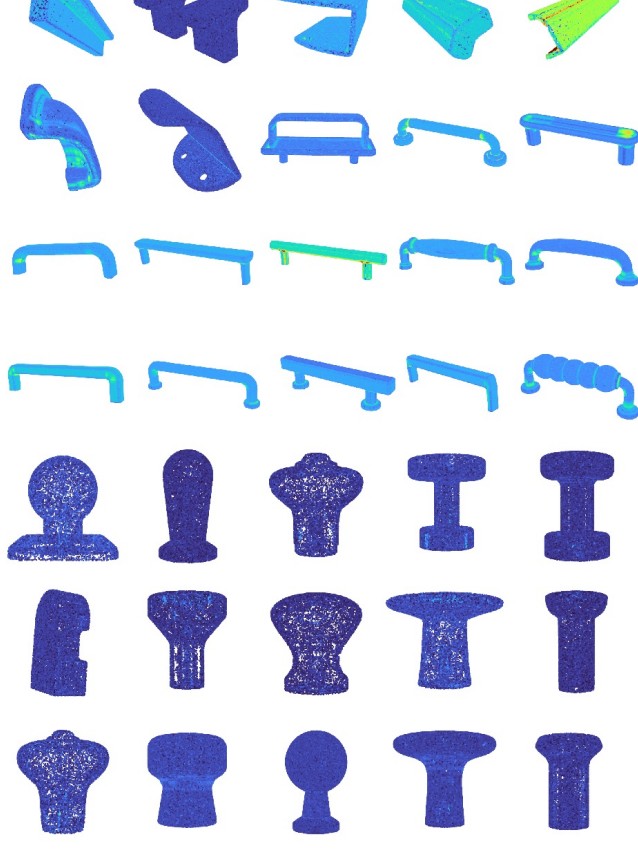

Fig. 11: Heatmap visualization of physical interaction fidelity ($H_{Q \to P}$) for Real-IKEA components. Dark blue indicates high precision. Crucially, the internal cavities required for hooking strategies are preserved.

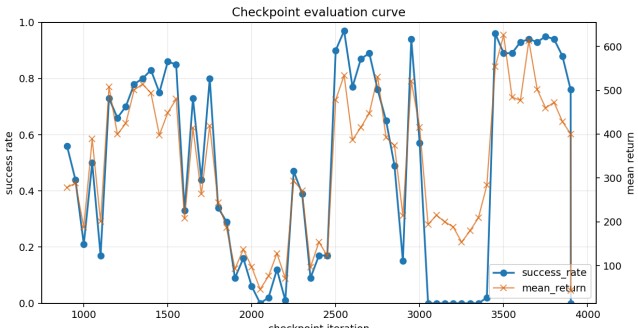

(a) Smooth Setting

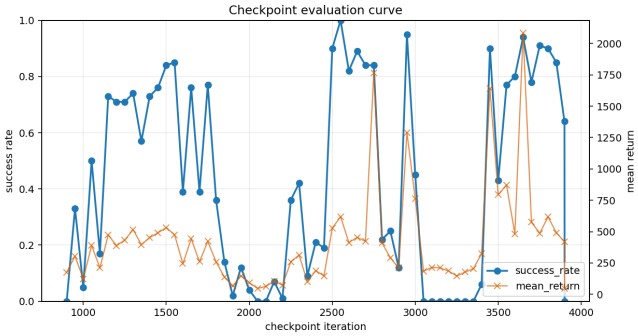

(b) Normal Setting

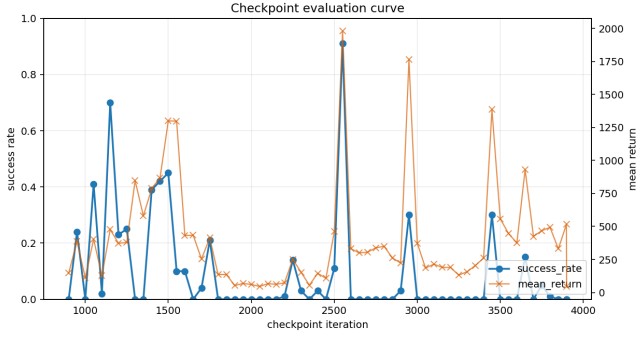

(c) High Resistance Setting

Fig. 12: Checkpoint evaluation curves across three distinct joint resistance settings. We report both the mean success rate (blue) and mean return (orange) for each saved iteration. Checkpoint 2550 consistently achieves near-perfect success rates across all three physical profiles.

domain randomization is fully activated to ensure the learned strategies generalize across spatial variations.

*7) Policy Selection and Robustness Evaluation:* Due to the highly dynamic nature of reinforcement learning and the aggressive domain randomization applied during the second phase of training, the policy's performance can fluctuate between iterations. To select the most robust Policy for deployment and downstream distillation, we systematically evaluated intermediate checkpoints saved during the training process.

We conducted a zero-shot evaluation of these checkpoints across three distinct joint resistance profiles to rigorously test their mechanical robustness:

- **Smooth:** low resistance (damping = 2, friction = 5);
- **Normal:** moderate resistance (damping = 5, friction = 10);
- **High Resistance:** severe resistance (damping = 10, friction = 20).

As illustrated in Figure 12, both the success rate and the mean return exhibit significant variance across iterations. Policies that overfit to specific dynamic parameters often collapse when tested under High Resistance. However, **Check-**

**point 2550** demonstrates exceptional robustness, achieving a success rate approaching 100% simultaneously across all three evaluation environments.

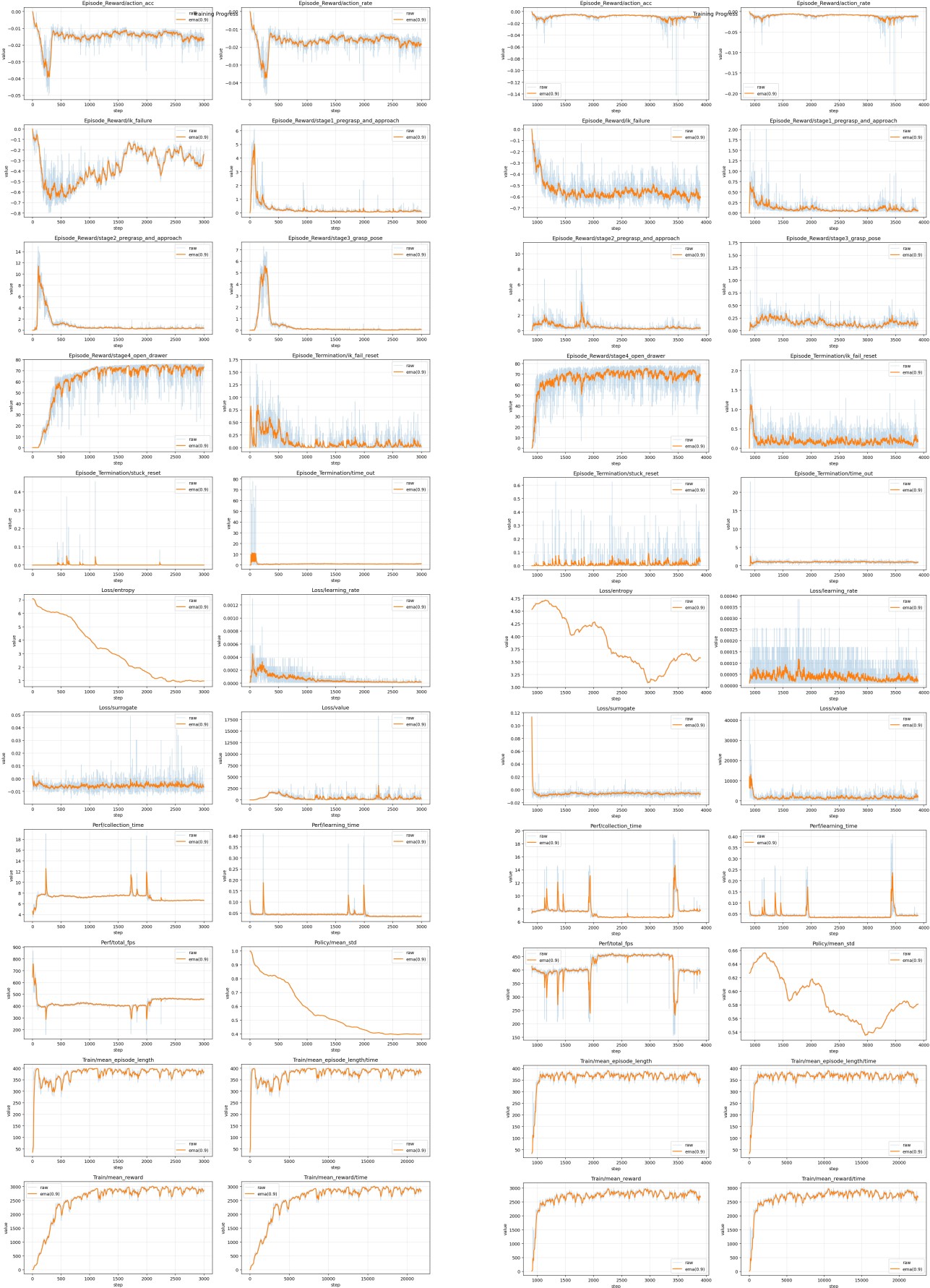

Fig. 13: Learning curves for the RL policy trained via PPO before activating domain randomization.

Fig. 14: Learning curves for the RL policy trained via PPO under domain randomization.

USE OF LARGE LANGUAGE MODELS

Large Language Models (LLMs) were used solely as writing assistants to help polish grammar and improve clarity of exposition. No content, technical claims, or experimental results were generated by LLMs. All scientific contributions and analyses are the work of the authors.