# OpenReview forum: "Real-IKEA: Why Physical Fidelity is the Prerequisite for Robust Manipulation"
_IEEE.org/ICRA/2026/Workshop/Manipulation_Robustness — ICRA 2026_

### Official Review · Reviewer_Lfhv · 2026-05-06
**A dataset and simulation framework of articulated object manipulation**

**Rating:** 7
**Confidence:** 3

**Review:**

Overall

This paper introduces Real-IKEA, a dataset and simulation framework built on MuJoCo, aimed at improving robustness in robotic manipulation by emphasizing physical fidelity. In particular, the work focuses on:

1) High-quality collision geometry via convex decomposition
2) Calibrated joint resistance (damping and friction)

The authors argue that current simulators bias policies toward fragile, friction-based strategies. They demonstrate that with higher-fidelity assets, RL agents can instead discover geometry-aware strategies, such as hooking and levering, which are more robust under resistance.

Strengths

The paper is well-motivated and clearly structured. The central premise—that physical fidelity plays a key role in robust manipulation—is intuitive and relevant to the community. The dataset scale (1,079 articulated configurations derived from real IKEA components) is meaningful and has potential practical value as a benchmark. The appendix further reflects substantial engineering effort in dataset construction and simulation design. Overall, the paper addresses an important aspect of robustness in manipulation.

Weaknesses

The experimental evaluation is somewhat limited in scope. In particular, the absence of real-world robot experiments weakens the central claim regarding robustness, as it remains unclear how well the learned behaviors transfer beyond simulation. Additional empirical validation, especially in real-world settings or across a broader set of tasks, would strengthen the paper.

---

### Decision · Program_Chairs · 2026-05-21

Accept